# Contribution of P2X4 Receptors to CNS Function and Pathophysiology

**DOI:** 10.3390/ijms21155562

**Published:** 2020-08-03

**Authors:** Alejandro Montilla, Gilda Paloma Mata, Carlos Matute, Maria Domercq

**Affiliations:** Department of Neuroscience, University of the Basque Country (UPV/EHU), Achucarro Basque Center for Neuroscience and Instituto de Salud Carlos III, Centro de Investigación Biomédica en Red de Enfermedades Neurodegenerativas (CIBERNED), 48940 Leioa, Spain; alejandro.montilla@ehu.eus (A.M.); gmata005@ikasle.ehu.eus (G.P.M.); carlos.matute@ehu.eus (C.M.)

**Keywords:** purinergic P2X4 receptor, microglia, neuron, inflammation, CNS disease

## Abstract

The release and extracellular action of ATP are a widespread mechanism for cell-to-cell communication in living organisms through activation of P2X and P2Y receptors expressed at the cell surface of most tissues, including the nervous system. Among ionototropic receptors, P2X4 receptors have emerged in the last decade as a potential target for CNS disorders such as epilepsy, ischemia, chronic pain, anxiety, multiple sclerosis and neurodegenerative diseases. However, the role of P2X4 receptor in each pathology ranges from beneficial to detrimental, although the mechanisms are still mostly unknown. P2X4 is expressed at low levels in CNS cells including neurons and glial cells. In normal conditions, P2X4 activation contributes to synaptic transmission and synaptic plasticity. Importantly, one of the genes present in the transcriptional program of myeloid cell activation is P2X4. Microglial P2X4 upregulation, the P2X4^+^ state of microglia, seems to be common in most acute and chronic neurodegenerative diseases associated with inflammation. In this review, we summarize knowledge about the role of P2X4 receptors in the CNS physiology and discuss potential pitfalls and open questions about the therapeutic potential of blocking or potentiation of P2X4 for different pathologies.

## 1. Introduction

The concept of purinergic neurotransmission was first proposed in 1972 by George Burnstock [1]. It is today accepted that all CNS cells, neurons, oligodendrocytes, astrocytes and microglia express P2 and P1 receptors. ATP can be co-released with other neurotransmitters at synapses, and it can be release extrasynaptically by pannexin opening as well as by other mechanisms by damaged cells. ATP and adenosine are essential for cell signaling and cell communication in the CNS [2]. On the other hand, immune cells, in the absence of pathogens, sense the injury by recognizing the release of molecules that are normally located inside the cell, known as damage-associated molecular patterns (DAMPs) or “endogenous danger signals” [3]. ATP has been characterized as a danger signal implicated in innate and adaptive immunity, leading to a plethora of responses through its interaction with their purinergic P2 receptors [3,4]. P2 receptors can be further subdivided into metabotropic P2Y receptors (P2YRs) that are G-protein-coupled and ionotropic P2X receptors (P2XRs 1–7) that are nucleotide-gated ion channels [4]. Previous reviews have addressed the pharmacology of P2X4 and the role of this receptor in CNS signaling [5]. We here update the role of ionotropic P2X4R in CNS physiology and discuss the therapeutic potential of P2X4 receptor manipulation in CNS pathologies. P2X4 receptors form homomeric or heteromeric assemblies with P2X6, P2X7 and P2X1. It is rapidly activated by ATP, has high permeability to Ca^2+^, slowly desensitizes and has high affinity to ATP (detailed description of the molecular and pharmacological properties of P2X receptors can be found in [5,6,7]).

## 2. P2X4R Location in the CNS

P2X4 expression occurs in most cell types in the central and peripheral nervous system (CNS and PNS, respectively) of mice and humans. ATP is involved in neurotransmission via P2X4 receptors (Figure 1). Accordingly, P2X4R is present in neurons throughout the CNS including the olfactory bulb to hypothalamus or the cerebellum [8]. Immunohistochemistry analysis also showed its presence in GABAergic neurons of the striatum and substantia nigra [9]. In addition, P2X4 is expressed in retinal ganglion neurons [10], supraoptic neurons [11] or neurons in the paraventricular nucleus [12] and in dorsal horn neurons [13]. In the PNS, P2X4 are present in peripheral ganglia [14,15] and in both sympathetic and parasympathetic neurons. More recently, using BAC transgenic mice expressing tdTomato under the control of the P2X4 receptor gene (*P2rx4*), this receptor was found in sparse populations of tdTomato-positive neurons in most brain areas and in particular in the hypothalamic arcuate nucleus [16]. P2X4 receptors in this nucleus are mainly located in AgRP-NPY neurons, at presynaptic level, and in tanycytes, playing a role in feeding control [16].

Importantly, P2X4 is mainly expressed in microglial cells, in which this receptor is involved in a wide variety of functions both in physiological conditions and in response to spinal nerve injury in rats [17]. Consistent with that observation, P2X4 upregulation has been reported in other models involving microglial activation, such as spinal cord injury, animal models of neuropathic and inflammatory pain, ischemia, multiple sclerosis or glioma (reviewed in [4,5]). Thus, the P2X4^+^ state of microglia seems to be associated to the microglial activation occurring in different paradigms of neuroinflammation [4].

In addition to neurons and microglia, astrocytes from the rat nucleus accumbens, as well as astrocytes from rat primary cortical cultures express P2X4 [18]. Immunochemical studies also revealed the presence of the receptor in astrocytes from hippocampal regions [19] and from the brainstem [20]. However, there is no functional evidence using electrophysiology, of the presence of P2X4 in hippocampal astrocytes [21]. In turn, mice cortical astrocytes did not show P2X4 expression and no specific response was detected in ATP-induced currents in the presence of ivermectin, a P2X4 allosteric modulator [22].

Finally, protein-level presence of P2X4R was found in cultured rat oligodendrocyte progenitors [23]. However, as in astrocytes, patch clamp studies in slices did not reveal functional P2X4 receptors in oligodendrocytes [24].

## 3. Role of P2X4 Receptor in CNS Physiology

### 3.1. Role of P2X4 Receptor in Synaptic Transmission

Adenosine 5-triphosphate (ATP) was firstly proposed as a transmitter in 1972, in this way introducing the purinergic signaling concept [1]. The first evidence of ATP release in a depolarization-induced manner from a synaptosomal fraction came a few years later [25]. In this way, purinergic receptors may modulate or participate in the synaptic transmission. ATP is co-released with other neurotransmitters such as acetylcholine, glutamate and GABA and act as a co-transmitter at neuromuscular junction and at excitatory and inhibitory synapses [26,27]. In addition, ATP can be released by perisynaptic astrocytes cells, which modulates postsynaptic efficacy [28].

P2X receptors subunits are present both in pre- and postsynaptic sites. Thus, presynaptic P2X receptors elicit glutamate release in sensory neurons [29], whereby they may control intracellular calcium homeostasis [30], as P2X4 receptors show a high permeability for this cation [5]. Specifically, P2X4 are immunolabelled in axon terminals and dendrites of neurons in the olfactory bulb and spinal cord, suggesting a role of these receptor in sensory synapses [31]. Indeed, presynaptic heteromeric P2X receptors including P2X4 and P2X6 subunits regulate synaptic activity by controlling co-transmission of other molecules in rat cultured dorsal horn neurons [32]. In the hypothalamus, ATP induced somatic P2X4-inward currents in supraoptic nuclei neurons and increased the frequency of spontaneous glutamatergic and GABAergic synaptic currents through presynaptic (not postsynaptic) P2X2 and P2X4 receptors [33]. In addition, ATP release from microglia activates presynaptic P2X4 hippocampal mossy fibers projecting to CA3 neurons [34].

On the other hand, postsynaptic P2X4 are present in excitatory, glutamatergic synapses in cerebellum and hippocampus, as revealed by electron microscopy [35]. In the latter, these receptors may regulate synapse strength in mouse hippocampal CA1 neurons [36]. In particular, calcium entry through P2X4Rs in association with NMDA receptors can facilitate long-term potentiation in CA1 region, a feature potentiated by ivermectin [37]. Activation of P2X4 receptors can modulate neurotransmitter responses as it can cross-inhibit α3β4 nicotinic channels when co-expressed in *Xenopus* oocytes [38] and GABA(A) receptors in hypothalamic neurons [39]. Finally, P2X4 in microglia modulates indirectly synaptic transmission by contributing to the release of brain-derived neurotrophic factor (BDNF) from these cells [40].

### 3.2. Role of P2X4 Receptors in Glial Cells

The functional significance of P2X4 receptors in microglia has received much attention, as these cells expressed relatively high levels of these receptors [5,17]. As mentioned above, P2X4 expression is increased in different models of microglial activation. The shift of microglia towards a P2X4^+^ reactive state is known to be governed by the IRF8-IRF5 transcriptional axis [41]. These transcription factors drive microglial pro-inflammatory activation and neuroinflammation in neurodegenerative events [42]. P2X4 receptors mediate early microglia cell death during neuroinflammation [43], although the physiological or pathological significance of microglia cell death is mainly unknown. In this sense, it has been proposed that resolution of pro-inflammatory microglia activation requires cell death and repopulation by a regenerative phenotype, which promotes regeneration in animal models of MS [44]. On the other hand, P2X4 activation regulates the inflammasome after spinal cord injury in other neural cells, in a coordinated action with pannexin-1 and P2X7 channels [45].

Microglia depend on their migration capacity and motility to perform an active surveillance of the brain. P2X4 activation drives microglia motility via the phosphatidylinositol-3-kinase (PI3K)/Akt pathway [46]. Thus, microglia moving towards injury areas express high levels of P2X4 receptor [4]. Interestingly, IRF8 is also necessary to mediate microglia/macrophage migration towards the epicenter of spinal cord lesions and to promote its recovery [47].

As mentioned above, microglia release BDNF through activation of P2X4 in spinal cord synapses [40]. In addition to its role in neurotransmission, BDNF enhances oligodendrocyte progenitor cells (OPC) differentiation to mature oligodendrocytes and regulates myelination via activation of tropomyosin-related kinase (Trk) B receptors [48]. Likewise, BDNF favors myelination in the peripheral nervous system [49] and remyelination after injury [50]. In addition to regulating the release of BDNF, P2X4 receptors are specifically involved in inflammatory-mediated prostaglandin E2 (PGE2) production, which contributes to pain-related inflammation [51]. Finally, P2X4 activation enhances innate immunity as it improves bacterial killing by macrophages and protects against sepsis [52].

Little is known about specific P2X4 roles in other glial cells such as astrocytes or oligodendrocytes. Interestingly, P2X4 activation regulates the release of BDNF in Schwann cells that, as in microglia, may promote functional recovery and remyelination in injury models [50].

### 3.3. Intracellular Role of P2X4 Receptor

In the healthy organism, P2X4 are constitutively internalized by the interaction between the adapter protein 2 (AP2) and a noncanonical endocytosis motif in the C-tail of P2X4 subunit [53]. As a result, P2X4 is targeted to endosomes, lysosomes and lysosome-related organelles, ensuring a low surface expression not only in neurons, but also in microglia and macrophages [54,55] and there is a dynamic regulation of P2X4 trafficking between the intracellular compartments and the plasma membrane [56]. In addition to regulating receptor density at the plasma membrane, and thus the response to extracellular ATP, these mechanisms may also accelerate recovery from desensitization [57], as internalization of P2X4 and trafficking to acidic compartments facilitate re-sensitization of P2X4 [58]. Indeed, blocking internalization by mutating trafficking motifs within the receptor or using an inhibitor of dynamin increases plasma membrane expression of P2X4R but slows the re-sensitization process of P2X4 [56,58]. Thus, recycling of P2X4 through the acidic organelles may regulate the cellular responsiveness in the sustained presence of ATP.

The pathological implications of P2X4 endocytosis blockage has been recently analyzed using a new genetic tool, conditional transgenic knock-in P2X4 mice (Floxed P2X4mCherryIN), allowing the Cre activity-dependent genetic swapping of the internalization motif of P2X4 by the fluorescent mCherry protein to prevent constitutive endocytosis of P2X4 [59]. These mice overexpress P2X4R at the surface of the targeted cells, mimicking the pathological increased surface P2X4^+^ state. Increased surface P2X4 density in the hippocampus of knock-in mice altered LTP and LTD plasticity phenomena at CA1 synapses without affecting basal excitatory transmission. Altogether, data suggest that the fine balance of intracellular and plasma membrane P2X4 is pivotal for CNS correct functioning.

The presence of P2X4 in lysosomes is essential not only to allow receptor membrane trafficking but also to control the function of lysosomes themselves, which has physiological and pathological implications. At lysosomes, P2X4 is oriented with their extracellular domain facing the acidic luminal environment [60]. Patch clamp studies of enlarged vacuolar lysosomes have revealed that lysosomal P2X4 are under dual regulation by luminal ATP and pH [61]. In the intracellular compartment, P2X4 is involved in vacuolation and lysosome trafficking as well as in endolysosomal Ca^2+^ release and membrane fusion, in a calmodulin-dependent manner [55]. Ca^2+^ release into the cytoplasm may subsequently generate diverse intracellular signaling [60,61].

Relevance of the function of lysosomal P2X4 emerge from data in different models of pathologies. P2X4 activation with the allosteric modulator ivermectin induces lysosomal acidification and potentiates myelin engulfment and degradation [24]. Thus, it is possible that these strategically located P2X4Rs could directly modulate myelin phagocytosis, an essential step for an efficient remyelination in MS [4]. In addition, activation of P2X4 triggers fusion of lysosomes with other intracellular vesicles to drive secretory processes [60]. In this sense, activation of lysosomal enriched P2X4 in Schwann cells enhanced the secretion of BDNF that promotes regeneration after injury [50].

In addition to lysosomes, P2X4 is expressed in lamellar bodies (LBs), which are large lysosome-related organelles storing pulmonary surfactant. Importantly, activation of P2X4 located at LBs is essential for LBs exocytosis, which determines plasma membrane expression of P2X4 and the release of surfactant to the extracellular space [62].

## 4. Role of P2X4 Receptor in CNS Pathologies

Despite the general overexpression of P2X4Rs in most neurological conditions, acute and chronic [4,63], the role played in the inflammatory cascades and in secondary brain damage could be different in each condition. Indeed, P2X4 receptor has a dual role in the inflammatory cascade of brain pathologies and thus future therapies range from potentiation of the receptor to its blockage. In section, we review data supporting the role of P2X4 in the etiology of CNS disorders and the rationale for therapies based on P2X4 receptor modulation (see summary in Table 1).

### 4.1. Spinal Cord and Peripheral Nerve Injury: Neuropathic Pain

Spinal cord injury (SCI) is one of the major causes of disability around the world, which causes irreversible loss of motor and sensory function as a result of axonal damage, demyelination and death of oligodendrocytes, astrocytes and neurons. In the post-acute phase, spontaneous tissue repair such as remyelination and axonal regeneration also occurs with partial motor recovery.

P2X4 are highly upregulated after spinal cord and peripheral nerve injury [17,40] as well as after traumatic brain injury [76]. Upregulation of P2X4 occurs mainly in microglia but also in neurons and Schwann cells [50]. The extracellular concentration of fibronectin after peripheral nerve injury (PNI) is pivotal in P2X4 upregulation [77]. Fibronectin increases P2X4 mRNA and protein expression through two fibronectin/integrin signaling pathways with Lyn of Src-family kinases as the starting point [78]. In addition, the interferon regulatory factors IRF5 and IRF8, which control immune system regulation, are key in shifting spinal cord microglia toward a P2X4-expressing reactive state after PNI [41]. IRF5 upregulates P2X4 expression by directly binding to the promoter region of the *P2RX4* gene [41]. Moreover, both signaling pathways seems to be related since fibronectin stimulates the translocation of IRF5 from the cytoplasm into the microglial nucleus [41].

In general, the impact of P2X4 activation after SCI or PNI on damage and/or recovery yielded contradictory results. After SCI, mice deficient in P2X4R showed impaired inflammasome signaling in the cord, resulting in decreased levels of IL-1β and reduced infiltration of neutrophils and monocyte-derived M1 macrophages, which in turn resulted in significant tissue sparing and improvement in functional outcomes [45]. In contrast, after sciatic nerve crush induction, a classic model to study peripheral nerve regeneration, overexpression of P2X4 in Schwann cells by genetic manipulation enhanced the secretion of BDNF, accelerated remyelination of injured axons and promoted motor and sensory functional recovery [50]. The first study attributed the effect to P2X4R stimulation in neurons, whereas the latter is due to P2X4 overexpression in Schwann cells. More studies are needed to clarify the cell specific roles of P2X4 in these disorders and to determine its therapeutic value to promote recovery. For instance, macrophages lacking interferon regulatory factor 8 (IRF8), a key factor in controlling P2X4 expression, cannot migrate toward the epicenter of the lesion after SCI, which determines inefficient remyelination and poor functional outcome [47].

One of the major symptoms of patients after suffering an acute injury is neuropathic pain, a form of chronic pain evoked by the damage of neurons involved in pain signaling that is refractory to nonsteroidal anti-inflammatory drugs and opioids. P2X4 are key in neuropathic pain after PNI and SCI [17,41]. A prominent symptom in neuropathic pain is abnormal pain hypersensitivity evoked by normally innocuous stimuli, known as mechanical allodynia. P2X4 contribute to mechanical allodynia [17]. Mechanistically, activation of P2X4 in spinal cord microglia induces the secretion of brain-derived neurotrophic factor (BDNF), causing an altered transmembrane gradient of Cl^−^ in a subpopulation of dorsal horn lamina I neurons, presumably through the downregulation of the neuronal chloride transporter KCC2 [79]. This in turn reverses in these neurons the polarity of GABA and glycine to depolarization instead of hyperpolarization, resulting in a higher excitability that causes spontaneous pain, hyperalgesia and tactile allodynia. In addition, BDNF released by microglia in response to P2X4 stimulation also leads to the phosphorylation of the NR1 subunit of NMDA receptors in dorsal horn neurons of the spinal cord [40]. In turn, P2X4 in sensory neurons in dorsal root ganglion neurons are coupled to BDNF-dependent signaling pathways, phosphorylation of Erk1/2 and GluN1 subunit and the downregulation of the co-transporter KCC2 in inflammatory pain conditions [80].

All these findings suggest that P2X4 blockers could be a promising therapeutic approach to alleviate neuropathic pain [63]. This includes antidepressants paroxetine and duloxetine that alleviated neuropathic pain [64] as well as NP-1815-PX that produces anti-allodynic effects in chronic pain models without altering acute pain sensitivity [65].

### 4.2. Epilepsy

Epilepsy, a major neurological disorder characterized by seizures, is associated to ATP release. ATP is released during seizures, either from neurons or from astrocytes [81], and rapidly signals to microglia inducing a large purinergic inward current that was mostly blocked in the presence of BBG [82]. Although this pharmacology suggests the main involvement of microglial P2X7, further pharmacological analysis should determine whether P2X4 receptors contribute to microglia responses after epilepsy-induced ATP release.

Changes in the expression of P2X receptors occur in various animal models of epilepsy. Thus, kainate (KA)-induced status epilepticus (SE) increases P2X4 expression in the hippocampus [83], mostly in microglia [84]. Accordingly, purinergic inward currents in microglia increased after status epilepticus [83] and intrahippocampal injection of KA induced an increase in P2X4 receptor mRNA expression in FACS isolated microglia [82]. However, other epilepsy models, such as intraamygdala KA and pilocarpine-induced SE, did not reveal changes in P2X4 expression [84,85].

Regarding the impact of P2X4 in the etiology of epilepsy, the only data available are based on the use of P2X4-deficient mice [66]. Purinergic receptor blockage did not alter the frequency or amplitude of the epileptic discharges in vitro [82] and the susceptibility to kainate-induced seizures did not change in P2X4-deficient mice [66]. However, microglia responses to KA-induced SE drastically change in P2X4-deficient mice as well as microglia recruitment, but not proliferation [66]. Importantly, P2X4 deletion blocked the increase of delayed rectifying outward potassium currents in microglia, a very interesting finding since Kv1.3 channels contribute to neurotoxic potential of microglia [86]. This feature may explain the decrease in neuronal death observed in P2X4-deficient mice after KA-induced SE [66].

### 4.3. Ischemia

Ischemic brain injuries resulting from a reduction in cerebral blood flow (CBF) causes severe disability and death. Pharmacological interventions are either ineffective and are limited by adverse effects, and neuroprotection in patients after ischemic brain damage consequently remains a major unfulfilled medical need.

As in other acute disease models, P2X4 receptors are upregulated in microglia or infiltrated macrophages [87,88] and their activation modulates the inflammatory response after stroke [67]. In a rat model of preterm hypoxia-ischemia, the expression of P2X4 was significantly increased in microglia and minocycline, a potent inhibitor of inflammation, attenuated the upregulation of P2X4Rs induced by hypoxia-ischemia [89]. In addition, increased expression of P2X4 was also observed in the hippocampus of gerbils subjected to bilateral common carotid occlusion [90]. Importantly, specific deletion of P2X4 in myeloid cells leads to a further exacerbation of the inflammatory reaction, a higher release of pro-inflammatory cytokines and a reduced expression of BDNF, which determines the lack of recovery and depressive phenotype after chronic stroke [67].

Ischemia induces ATP release and activation of purinergic receptors, such as P2X7, that contribute to the anoxic inward current [91]. However, the contribution of P2X4 to the anoxic current is unknown. P2X4 antagonists do not modify the size of the infarct or the neurological deficits after middle cerebral artery occlusion (MCAO) [68]. These results are in contrast with those in another study showing a reduction of infarct volume in global P2X4 KO mice [67]. Beneficial effects of blocking P2X4 has also been described in a model of neonatal hypoxia in rat using TNP-ATP antagonist [69].

Importantly, P2X4 is also abundantly expressed in vascular endothelial cells [92,93], where it acts as a mechanoreceptor to sense blood flow changes [93]. Ischemic preconditioning (IPC), defined as transient ischemia and subsequent reperfusion, exerts neuroprotective effects against lethal ischemia. As shear stress sensing in blood is pivotal in IPC, Ozaki and colleagues observed that blockade of P2X4 with BDBD abolished the ability of IPC to prevent infarct formation and neurological impairment after prolonged ischemia. Mechanistically, the activation of P2X4 receptors in vascular endothelial cells is linked to an increased osteopontin expression leading to neuroprotection [68].

### 4.4. Multiple Sclerosis

Multiple sclerosis (MS) is a demyelinating disease of the central nervous system (CNS) characterized by an autoimmune inflammatory reaction to myelin components leading to oligodendrocyte death and demyelination and axonal damage [94]. Demyelination in focal lesions or plaques in the white and grey matter of the brain and spinal cord characterizes the initial phase of the disease. MS is one of the few diseases or insults into the CNS in which there is a partial or total regeneration although the remyelination capacity declines with age and disease evolution [94].

P2X is overexpressed in parenchymal microglia and in inflammatory foci in acute and chronic animal models of the disease, encephalitis autoimmune experimental (EAE) [43,95] and multiple sclerosis samples [43]. Activation of P2X4 is beneficial for EAE recovery. Thus, P2X4 blockage exacerbate the neurological symptoms in EAE whereas P2X4 potentiation with ivermectin ameliorates motor symptoms in EAE. Pharmacological data were further assessed using the P2X4 deficient mice [24]. Although P2X4 receptors in myeloid cells appear to drive these effects, further experiments using cell specific KO mice would be necessary to corroborate this claim. Antagonist or the positive modulator ivermectin did not alter immune priming in EAE mice [24], despite the fact that P2X4 could modulate T cell activation and migration [96].

The benefits of macrophages in MS are attributed to being required in clearing myelin debris after a demyelinating episode by phagocytosis [97] as well as their release of a variety of growth factors into the injured CNS that favor the OPC remyelination program [94]. In microglia, P2X4 blockage favors a shift towards a more proinflammatory-phenotype that inhibits myelin phagocytosis and halts oligodendrocyte differentiation [24]. P2X4 activation in microglial cells induce BDNF release [79], a growth factor that promotes oligodendrocyte differentiation. Both Schwann cells and microglia production of BDNF secondary to P2X4 potentiation or overexpression promote remyelination [24,50]. On the other hand, myelin debris contains inhibitors of OPC differentiation that stop remyelination [94], and thus macrophage myelin phagocytosis capacity, which depends on age and macrophage activation state [97], is pivotal for efficient remyelination. As P2X4 potentiation by ivermectin increases myelin engulfment and degradation [24], this drug is a potential candidate to promote the repair of myelin damage in MS.

### 4.5. Neurodegenerative Diseases

The role of P2X4 receptors has not been thoroughly challenged in models of neurodegenerative disorders despite the altered phenotype observed in P2X4 deficient mice that exhibit sensorimotor deficits and behavior disturbances [72,98]. Those deficits are attributed to altered dopamine signaling in P2X4 KO mice [70] and calls the attention for a role of these receptors in dopamine homeostasis and in motor control and sensorimotor gating. Indeed, in a model of Parkinson’s disease using 6-hydroxydopamine for dopamine depletion, P2X4 KO mice exhibited an attenuated levodopa induced motor behavior, whereas ivermectin enhanced this behavior [70]. These data indicate that P2X4 are relevant in maintaining dopamine homeostasis as well as associated behaviors, and, therefore, it may contribute to the pathophysiology of Parkinson’s disease. However, their role of P2X4 in this disorder is still practically unexplored.

Increased P2X4 expression and surface density in neurons occurs in the hippocampus of Alzheimer’s disease (AD) patients with severe cognitive impairment [71], suggesting that upregulated P2X4 may contribute to synaptic dysfunction. Thus, overexpression of P2X4 in neurons enhanced the toxic effect of amyloid Aβ), while silencing of this purinergic receptor decreased cellular death after exposure to Aβ [71]. In addition, Aβ induced a caspase-3-mediated cleavage of the receptor that slowed channel closure times and prevented agonist-induced internalization of the receptor [71]. On the other hand, rats expressing mutated human superoxide dismutase (mSOD1G93A), an animal model for amyotrophic lateral sclerosis (ALS), showed a strong P2X4 immunoreactivity selectively associated with degenerating motoneurons in spinal cord and cerebral cortex as well as in Purkinje cells of the cerebellar cortex. These neurons did not display apoptotic markers but showed other signs of abnormality, such as loss of the neuronal marker NeuN and recruitment of microglial cells with neuronophagic activity [99]. Altogether, these findings strongly suggest that abnormal trafficking and proteolytic processing of the P2X4 protein may contribute to neuronal cell death in neurodegenerative diseases. This idea is further supported by data obtained using a conditional knock-in mice (P2X4mCherryIN) mimicking the pathological increase of surface P2X4 [59]. The increased expression of P2X4R at the surface of excitatory neurons decreases anxiety, impairs memory processing and alters activity-dependent synaptic plasticity phenomena in the hippocampus suggesting that upregulation of neuronal P2X4 observed in AD [71] may have key roles in AD pathogenesis [59].

### 4.6. P2X4 and Psychiatric Diseases

Only a few studies associate P2X4 receptors to psychiatric disorders and their animal models. Prepulse inhibition (PPI) of acoustic startle reflex is a robust operational measure of sensorimotor gating, which is a widely used readout in experimental and clinical psychiatry as it is deficient in schizophrenia and bipolar disorder [100]. PPI is defined as attenuation of behavioral response to a strong sensory stimulus (pulse) when preceded by a weaker stimulus (prepulse). The observed reduction in response to the pulse stimulus occurs due to activation of inhibitory mechanisms in the CNS. Hence, PPI serves as a mechanism to avoid interference between distinguishable stimuli. The inability to avoid this interference results in significant inundation or overflow of incoming sensory information leading to impairments in attention-dependent cognitive functions [73]. Deficits in PPI have been reported in a wide spectrum of neuropsychiatric disorders that are characterized by cognitive deficits including schizophrenia and autism spectrum disorders. Mice deficient in P2X4 exhibited PPI deficits that were alleviated by dopamine (DA) receptor antagonists demonstrating an interaction between P2X4 and DA receptors in PPI regulation [70,73]. In addition, although P2X4 deficiency did not induce significant alterations of locomotor activity and anxiety-related indices, it induced enhanced tactile sensitivity and significant reductions in social interaction and maternal separation-induced ultrasonic vocalizations in pups [72]. These studies highlight a putative role of P2X4 in the regulation of perceptual and sociocommunicative functions and point to these receptors as putative targets for disturbances associated with neurodevelopmental disorders.

In addition, P2X4 has been proposed as a target for therapeutic interventions in alcohol use disorders (AUDs) [74]. P2X4 is expressed in neurons as well as in glial cells in the reward circuitry of the mammalian brain, the mesocorticolimbic pathway, where it plays a role in synaptic plasticity [35,40,66]. On the other hand, there are seminal findings that supports a role of P2X4 in alcohol use disorders. First, P2X4 activity is selectively blocked in the presence of ethanol, which acts as a negative allosteric modulator [101,102]. Second, mice lacking the *p2rx4* gene drink more EtOH than wildtype controls [74]. On the other hand, ivermectin and other avermectins antagonize the inhibitory effect of ethanol in vitro and reduce EtOH intake and preference in vivo [75]. Overall, these findings suggest that alcohol intake may be modulated by ethanol acting on P2X4 and that pharmacological potentiation of P2X4 by ivermectin or avermectin analogs may reduce alcohol consumption and preference.

## 5. Conclusions

P2X4 receptors are abundantly expressed in different cell types and regions in the CNS. This widespread distribution and their higher affinity for ATP, in comparison to P2X7, determines that P2X4 actively participate in synaptic transmission and different key CNS functions. This fact determines their wide impact on CNS disorders and their therapeutic potential. Development of specific pharmacological tools to selectively target P2X4 receptors will help to corroborate their physiological and pathological in the CNS. In addition, cell-specific P2X4 knockout mice will shed light on the complex mechanism involved in each pathological situation.

## Figures and Tables

**Figure 1 ijms-21-05562-f001:**
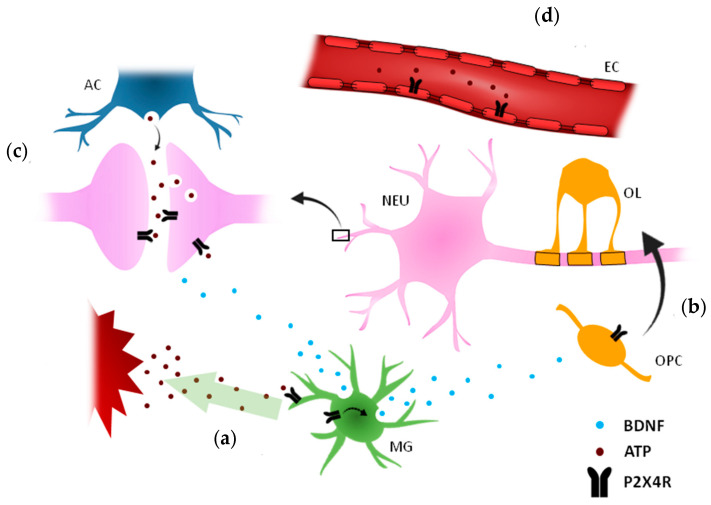
Summary of the specific roles played by P2X4 receptors in each central nervous system (CNS) cell subtypes. (**a**) ATP is released in response to acute injury and P2X4 plays a pivotal role on microglia chemotaxis and motility. In addition, microglia P2X4 activation induces brain-derived neurotrophic factor (BDNF) production and secretion, which modulates synaptic efficacy and accelerates oligodendrocyte progenitor cell (OPC) differentiation to mature oligodendrocytes and thus remyelination. (**b**) OPCs cells express P2X4, although its capacity to respond to ATP or its potential role in OPC differentiation remains to be defined. (**c**) ATP, released by neurons as a co-transmitter or released by astrocytic processes surrounding synapses, activates neuronal P2X4 located at pre- and postsynaptic sites and modulates neurotransmitter release or postsynaptic efficacy. (**d**) Finally, P2X4 receptors are expressed by endothelial cells in the luminal side of the vessel. Importantly, P2X4 modulates endothelial response to changes in blood pressure.

**Table 1 ijms-21-05562-t001:** Therapeutic potential of P2X4 modulators, antagonists or allosteric modulators in CNS pathologies.

Pathology	Model	Therapeutic Benefit	Ref.
Neuropathic and inflammatory pain	Peripheral nerve or spinal cord injury in mice	Antisense oligonucleotide to P2X4 alleviates neuropathic pain	[17]
		P2X4^−/−^ mice lack mechanical hyperalgesia and have reduced inflammatory pain	[40,51]
		Neuropathic pain was alleviated in the present of P2X4 antagonists such as paroxetine, duloxetine, NP-1815-PX	[64,65]
Spinal cord and peripheral nerve injury	Spinal cord injury in mice	P2X4R^−/−^ mice showed impaired inflammasome signaling and improved functional outcome	[45]
	Sciatic nerve crush in mice	Overexpression of P2X4R promoted motor and sensory functional recovery	[50]
Epilepsy	Kainate induced status epilepticus in mice	P2X4^−/−^ mice showed ameliorated microglia response and reduced neuronal death	[66]
Ischemia	Middle cerebral artery occlusion (MCAO) (60 min) in mice	P2X4^−/−^ mice showed reduced infarct volume	[67]
	MCAO in mice	P2X4R antagonists did not affect MCAO-mediated infarct formation	[68]
	Neonatal hypoxia in mice	TNP-ATP antagonist reduced hypomyelination and cognitive decline	[69]
Multiple sclerosis	Experimental autoimmune encephalomyelitis	Ivermectin ameliorate clinical signs	[24]
	Lysolecithin (LPC) model in organotypic slices	Ivermectin potentiates remyelination after in LPC-induced demyelination	[24]
Parkinson’s disease	6-hydroxydopamine model of DA depletion	P2X4R KO mice exhibited an attenuated levodopa (L-DOPA)-induced motor behavior, whereas ivermectin enhanced this behavior	[70]
Alzheimer’s disease	Amyloid β (Aβ) exposure in vitro	Overexpression of P2X4Rs in neurons enhanced Aβ toxicity while silencing of P2X4Rs decreased neuronal death	[71]
	Conditional knock-in mice (P2X4mCherryIN) mimicking the pathological increase of surface P2X4R	Impairment of memory processing and altered synaptic plasticity in the hippocampus	[59]
Psychiatric disorders	P2X4R KO mice	P2X4 KO mice showed altered prepulse inhibition and reductions in social interactions	[70,72,73]
	Alcohol use disorders models in mice	Ivermectin and other avermectins reduces EtOH intake and preference	[74,75]
	Anxiety	Increases surface expression of P2X4 at excitatory synapses alleviates anxiety	[59]

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
