# Peer review of "Contribution of P2X4 Receptors to CNS Function and Pathophysiology"

_ijms, 2020, doi:10.3390/ijms21155562_

Round 1

Reviewer 1 Report

This is a concise and helpful review, which summarizes our  knowledge on P2X4 functions in the CNS. The authors sum up the distribution, function of P2X4 in physiological and pathological status.

Minor comments :

  1. According to UPHAR nomenclature, P2X receptors should be written P2X and not P2XR. Please replace P2X4R or P2X7R by P2X4 or P2X7  in the  manuscript as well as for other P2X subtypes.
  2. P1, l40. “similar to P2X7R” is it necessary?
  3. Paragraph 3.1. L99-106. A few important roles of P2X4 are missing. The notion of ATP as cotransmitter co-released with glutamate at excitatory synapses and GABA at inhibitory synapses (Jo and schlichter Nat Neurosci1999, Jo and Role JNeurosci 2002) should be mentioned along with the role of postsynaptic P2X4.
  4. ATP is also released by astrocytes (Gordon et al  Nature Neurosci 2005) and activation of  postsynaptic P2X2 or P2X4 by glial ATP in the hippocampus was showed to decrease synaptic strength by altering the nmber of synaptic AMPAR (Pougnet al al; Neuron 2014 and Pougnet et al Scientific Rep 2016 ).
  5. L314-315 correct font size
  6. Table:1 Amyotrophic lateral sclerosis and anxiety are both evoked in the text and should be included in the table with corresponding references. (anxiety may be in psychiatric disorders?)
  7. L228-229. please clarify. This notion seems to be linked to neuropathic pain.  But Lalisse et al described the increase of expression of P2X4 in sensory neurons in dorsal root ganglia  (also please precise this point) and subsequent signaling cascade  in chronic inflammatory pain conditions and not in neuropathic pain.
  8. L71-72 legend fig1: please correct typos “released by astrocityc processes surrounding synpases”.
  9. Precised ATP, released by neurons  as a co-transmitter or….

Author Response

We would like to submit the revised version of the manuscript ijms-868031 “Contribution of P2X4 receptors to CNS function and pathophysiology” by A. Montilla, GP Mata, C Matute and myself for publication as a reviewer in your journal IJMS.

We have addressed all the concerns and comments raised by the reviewers. All the minor points have also been corrected. All changes are highlighted in yellow.

Below you can see the detailed point-by-point response to the reviewers’ comments. We highly appreciate their helpful suggestions and criticisms. We thank you for considering our revised manuscript and we hope that it is now acceptable for publication in IJMS.

Sincerely,

Maria Domercq

Reviewer 1

  1. According to UPHAR nomenclature, P2X receptors shouldbe written P2X and not P2XR. Please replace P2X4R or P2X7R by P2X4 or P2X7 in the manuscript as well as forother P2X subtypes.

We have replaced them.

  1. P1, l40. “similar to P2X7R” is it necessary?

It has been eliminated

  1. L99-106. A few important roles of P2X4are missing. The notion of ATP as cotransmitter co-released with glutamate at excitatory synapses and GABA at inhibitory synapses (Jo and schlichter NatNeurosci1999, Jo and Role JNeurosci 2002) should be mentioned along with the role of postsynaptic P2X4.

We agree with the reviewer and we have included the concept in the revised version, lines 91-94.

  1. ATP is also released by astrocytes (Gordon et al NatureNeurosci 2005) and activation of postsynaptic P2X2 or P2X4 by glial ATP in the hippocampus was showed to decrease synaptic strength by altering the number of synaptic AMPAR (Pougnet al al; Neuron 2014 andPougnet et al Scientific Rep 2016 ).

The concept of ATP released by astrocytes has been included in lines 94-96. Regarding P2X2-dependent modulation of synaptic strength (Pougnet al al; Neuron 2014 and Pougnet et al Scientific Rep 2016 ), we have not added these data because the review is focused on P2X4.

  1. L314-315 correct font size

Corrected

  1. Table:1 Amyotrophic lateral sclerosis and anxiety are both evoked in the text and should be included in the table with corresponding references. (anxiety may be in psychiatricdisorders?)

We have included in the table only those pathologies in which P2X4 manipulation has an impact on the clinical symptoms or the pathological mechanisms. There is now sufficient data for ALS in our opinion but we have included anxiety in the table.

  1. L228-229. please clarify. This notion seems to be linked to neuropathic pain. But Lalisse et al described the increase of expression of P2X4 in sensory neurons in dorsal root ganglia (also please precise this point) and subsequent signaling cascade in chronic inflammatory pain conditions and not in neuropathic pain.

We have clarified  this point, lines 236-239.

  1. L71-72 legend fig1: please correct typos “released by astrocityc processes surrounding synpases”.

Corrected

  1. Precised ATP, released by neurons as a co-transmitteror...

Corrected

Reviewer 2 Report

In the present review, Maria Domerq and colleagues discuss the role of neuronal and glial P2X4 receptor in central as well as in peripheral nervous system. First, the authors describe the expression and function of P2X4 receptor in neurons and glia cells, involvement in synaptic transmission, and changes in membrane expression or cellular distribution. Then, they discuss the role of this receptors in PNS and CNS pathologies, with a specific focus on the role in spinal cord and peripheral nerve injury, neuropathic pain, epilepsy, ischemia, multiple sclerosis, neurodegenerative diseases and psychiatric diseases. This last part allows to highlight a potential role of P2X4 modulators or antagonists in these pathologies. This part is certainly interesting for a broad audience as it underlines why the physiology and pharmacology of P2X4 receptor and its expression in neuronal cells might be important.

The authors covered well the fields of P2X4 receptor and neuronal physiology and pathophysiology (97 references). This group has obviously an expertise in the field, as they published several articles on the role of P2X4 receptors in microglia (Abiega et al, 2016) and multiple sclerosis (Domercq et al., 2019).

In conclusion, the review is clear, well written, well documented and the illustration in figure 1 and literature summary in Table 1 are excellent. I think this review can be interesting for a broad audience and therefore deserves to be published in a good journal such as IJMS.

Minor comments:

Line 41: Ref [5], Stokes et al, 2017, should be mentioned in this article elsewhere, as the most recent previous review focused on similar subject, the role of P2X4 in nervous system, and the authors should emphasize what was added in their present review. In connection with basal properties of P2X4 (lines 40-41), some more original, seminal, articles summarizing molecular properties of P2X receptors including P2X4, for example North, Physiol Rev 2002 or Coddou et al., Pharmacol Rev, 2011, should be cited.

Line 43: should be “.. in the CNS“

Line 45: "In particular.....neuroinflammation [4]". I suggest to transfer this paragraph related to microglia after next paragraph describing expression of P2X4 receptor in neurons. Neurons are the most important cells in the CNS, like this it seems that microglia are crucial.

Line 72: “However, its impact is practically unknown”. It is not completely true, because it is already well established that presynaptic P2X4 receptors play a modulatory role, potentiating neurotransmitter release (as mentioned by the authors latter in this article.

Line 100: Study by Rubio and Soto, 2001, Ref [29], suggests that P2X4 is located postsynaptically based on immunogold labeling. However, functional evidence for the expression of P2X4 receptors, as well as other subtypes of P2X receptors, in postsynaptic membrane of CNS neurons are lacking, and no postsynaptic currents mediated by P2X4 receptor are documented  (see for example in hypothalamus, Vavra et al, Neuroscience, 2011).

Line 133: At least some summary text about the role of P2X4 in astrocytes or oligodendrocytes should be included, as these cells are mentioned in illustration on Fig. 1. I tis not sufficient to say that „little is known...“.

Line 282: observed

Line 333: please correct following expression „amyloid αβ)“

Line 383: „In addition, cell-specific P2X4 knockout mice will shed light...“ Do such mice already exist? It would be interesting to include some examples.

Author Response

We would like to submit the revised version of the manuscript ijms-868031 “Contribution of P2X4 receptors to CNS function and pathophysiology” by A. Montilla, GP Mata, C Matute and myself for publication as a reviewer in your journal IJMS.

We have addressed all the concerns and comments raised by the reviewers. All the minor points have also been corrected. All changes are highlighted in yellow.

Below you can see the detailed point-by-point response to the reviewers’ comments. We highly appreciate their helpful suggestions and criticisms. We thank you for considering our revised manuscript and we hope that it is now acceptable for publication in IJMS.

Sincerely,

Maria Domercq

Reviewer 2

Line 41: Ref [5], Stokes et al, 2017, should be mentioned in this article elsewhere, as the most recent previous review focused on similar subject, the role of P2X4 in nervous system, and the authors should emphasize what was added in their present review.

In connection with basal properties of P2X4 (lines 40-41), some more original, seminal, articles summarizing molecular properties of P2X receptors includingP2X4, for example North, Physiol Rev 2002 or Coddou et al.,Pharmacol Rev, 2011, should be cited.

We have cited these reviews and mentioned Stokes et al elsewhere.

Line 43: should be “.. in the CNS“

Corrected

Line 45: "In particular.....neuroinflammation [4]". I suggest to transfer this paragraph related to microglia after next paragraph describing expression of P2X4 receptor in neurons. Neurons are the most important cells in the CNS, like this it seems that microglia are crucial.

We have corrected the order

Line 72: “However, its impact is practically unknown”. It is not completely true, because it is already well established that presynaptic P2X4 receptors play a modulatory role, potentiating neurotransmitter release (as mentioned by the authors latter in this article.

We have changed this point.

Line 100: Study by Rubio and Soto, 2001, Ref [29], suggests that P2X4 is located postsynaptically based on immunogoldlabeling. However, functional evidence for the expression of P2X4 receptors, as well as other subtypes of P2X receptors, in postsynaptic membrane of CNS neurons are lacking, and no postsynaptic currents mediated by P2X4 receptor are documented (see for example in hypothalamus, Vavra et al,Neuroscience, 2011).

This point have been discussed in lines 104-107.

Line 133: At least some summary text about the role of P2X4in astrocytes or oligodendrocytes should be included, as these cells are mentioned in illustration on Fig. 1. I tis not sufficientto say that „little is known...“.

P2X4 appear in OPCs in figure 1, not in astrocytes or oligodendrocytes. We have included a potential role of P2X4 in OPCs.

Line 282: observed

Corrected

Line 333: please correct following expression „amyloid

Corrected